# The Impact on Survival and Morbidity of Portal–Mesenteric Resection During Pancreaticoduodenectomy for Pancreatic Head Adenocarcinoma: A Systematic Review and Meta-Analysis of Comparative Studies

**DOI:** 10.3390/cancers12071976

**Published:** 2020-07-20

**Authors:** Alessandro Fancellu, Niccolò Petrucciani, Alberto Porcu, Giulia Deiana, Valeria Sanna, Chiara Ninniri, Teresa Perra, Valentina Celoria, Giuseppe Nigri

**Affiliations:** 1Unit of General Surgery 2—Clinica Chirurgica, Department of Medical Surgical and Experimental Sciences, University of Sassari, V. le San Pietro 43, 07100 Sassari, Italy; alberto@uniss.it (A.P.); giulia.deiana2@gmail.com (G.D.); ninnirimariachiara@hotmail.it (C.N.); teresaperra92@tiscali.it (T.P.); valentinaceloria93@gmail.com (V.C.); 2Department of Medical and Surgical Sciences and Translational Medicine, St. Andrea University Hospital, Sapienza University of Rome, Via di Grottarossa 1037, 00189 Rome, Italy; nicpetrucciani@hotmail.it (N.P.); giuseppe.nigri@uniroma1.it (G.N.); 3Unit of Medical Oncology, AOU Sassari, Via E. De Nicola, 07100 Sassari, Italy; valeria.sanna@aousassari.it

**Keywords:** pancreaticoduodenectomy, portal-mesenteric resection, survival, complications

## Abstract

*Background:* The literature is conflicting regarding oncological outcome and morbidity associated to portal–mesenteric resection during pancreaticoduodenectomy (PD) in patients with pancreatic head adenocarcinoma (PHAC). *Methods:* A meta-analysis of studies comparing PD plus venous resection (PD+VR) and standard PD exclusively in patients with adenocarcinoma of the pancreatic head was conducted. *Results:* Twenty-three cohort studies were identified, which included 6037 patients, of which 28.6% underwent PD+VR and 71.4% underwent standard PD. Patients who received PD+VR had lower 1-year overall survival (OS) (odds radio OR 0.79, 95% CI 0.67–0.92, *p* = 0.003), 3-year OS (OR 0.72, 95% CI 0.59–0.87, *p* = 0.0006), and 5-year OS (OR 0.57, 95% CI 0.39–0.83, *p* = 0.003). Patients in the PD+VR group were more likely to have a larger tumor size (MD 3.87, 95% CI 1.75 to 5.99, *p* = 0.0003), positive lymph nodes (OR 1.24, 95% CI 1.06–1.45, *p* = 0.007), and R1 resection (OR 1.74, 95% CI 1.37–2.20, *p* < 0.0001). Thirty-day mortality was higher in the PD+VR group (OR 1.93, 95% CI 1.28–2.91, *p* = 0.002), while no differences between groups were observed in rates of total complications (OR 1.07, 95% CI, 0.81–1.41, *p* = 0.65)*. Conclusions:* Although PD+VR has significantly increased the resection rate in patients with PHAC, it has inferior survival outcomes and higher 30-day mortality when compared with standard PD, whereas postoperative morbidity rates are similar. Further research is needed to evaluate the role of PD+VR in the context of multimodality treatment of PHAC.

## 1. Introduction

Pancreatic head adenocarcinoma (PHAC) accounts for 80–90% of tumors of the exocrine pancreas and remains associated with a dismal long-term prognosis [1,2,3]. Pancreaticoduodenectomy (PD) with adjuvant chemotherapy offers potential cure for patients with PHAC, although even when surgical resection is considered curative, the five-year survival rate barely reaches 25% [4,5,6].

Involvement of the portal–mesenteric axis is quite frequent at the time of diagnosis of PHAC, being encountered in 25–30% of patients undergoing PD [7,8]. In particular, solid tumor contact with the portal vein or superior mesenteric vein of >180°, or contact of ≤180° with contour irregularity or thrombosis of the vein are patterns included in the category of borderline resectable tumors [9]. Since venous infiltration has not longer been considered a contraindication to resection, in the last decade, there has been a surge in the surgical treatment of borderline resectable PHAC, with the aim of obtaining R0 resection. This trend is also fueled by the encouraging results of systemic preoperative and postoperative treatments [10,11,12,13,14]. The use of PD with concurrent portal–mesenteric resection has in fact shifted the field of operable disease, and it is nowadays carried out in up to 20–30% of PD at high-volume pancreatic surgery centers [15,16].

The resection or replacement of the portal–mesenteric axis during PD represents a challenging and skill-demanding procedure leading to potential complications, in spite of debatable advantages in survival outcomes. Previous reviews investigating the impact of portal–mesenteric resection on the oncological outcomes following PD are limited and conflicting, since the results have predominantly been reported together with those of patients receiving distal or total pancreatectomy [6]. Other meta-analyses were limited by heterogeneity, because they also included tumor histotypes different from pancreatic adenocarcinoma, evaluated studies in whom arterial resection was performed along with venous resection [5], or did not evaluate survival outcomes [5,7,17].

Furthermore, none of previous reviews reported on preoperative and postoperative systemic treatment. Thus, from the present literature, it is difficult to extrapolate valuable conclusions on patients receiving PD with portal–mesenteric resection specifically for PHAC. In addition, newer and larger comparative studies have been added to the literature.

To overcome these limitations, we performed a systematic review and a meta-analysis limited to studies comparing PD with venous portal–mesenteric resection (PD+VR) versus standard PD in patients with PHAC only, in order to critically evaluate the effects on survival outcomes and postoperative complications.

## 2. Methods

### 2.1. Study Selection and Data Extraction

A systematic literature search using the PubMed, Web of Sciences (WOS), and Scopus databases was performed in December 2019 to identify studies reporting surgical outcomes of portal–superior mesenteric resection in patients undergoing PD for PHAC. The following keywords were used and combined for the search: ‘pancreaticoduodenectomy’, ‘duodenopancreatectomy’, ‘Whipple or Kausch–Whipple operation’, ‘venous resection’, ‘portal resection’, ‘mesenteric vein resection’, portal–mesenteric or porto-mesenteric resection’, ‘pancreatic carcinoma’, and ‘pancreatic head adenocarcinoma’. The ‘related articles’ function was used to broaden the search, and all abstracts, citations, and studies scanned as well as the references of relevant articles were reviewed (Appendix A). No language restrictions were made. Potentially relevant articles were examined by three independent investigators (A.F., G.D, V.C.) who extracted the following data: first author; year of publication; study design; number of subjects; patient and tumor characteristics; intraoperative outcomes; postoperative outcomes; survival outcomes.

### 2.2. Inclusion Criteria and Assessment of Study Quality

To be included in this meta-analysis, studies had to meet the following criteria: (1) comparing surgical outcomes and survival outcomes of PD with portal–mesenteric vein resection (PD+VR) and standard PD in patients with PHAC, (2) containing a previously unreported patient group. If a patient cohort was reported more than once by the same institution, the most informative and/or recent article was included in our analysis. Quality of the included studies was assessed according to the Newcastle–Ottawa Quality Assessment Scale (NOS) for cohort studies [18] (Appendix A).

### 2.3. Exclusion Criteria

We excluded from our meta-analysis studies comparing PD+VR and PD for the following: (a) tumors other than pancreatic adenocarcinoma (such as ampullary carcinoma, cholangiocarcinoma, primitive duodenal cancer, and malignant neuroendocrine tumors); (b) pancreatic cancer treated with total pancreatectomy, or distal pancreatectomy; (c) patients in whom also arterial resection was carried out along with portal–superior mesenteric resection. All those studies were evaluated and excluded only when it was impossible to extract the data specifically related to patients receiving PD+VR and PD for PHAC. We also excluded studies in which the outcomes of interest (specified below) were not reported or impossible to calculate for both PD+VR and PD groups (Appendix A).

### 2.4. Outcomes of Interest and Definitions

All the studies were abstracted for the following relevant data:Patients baseline characteristics: age, gender, American Society of Anesthesiologists Classification (ASA Class), number of patients receiving neoadjuvant chemotherapy, number of patients undergoing preoperative biliary drainage for obstructive jaundice.Tumor characteristics: stage according to AJCC, T stage according to the TNM, tumor size, grade of differentiation (well, moderate, and poor), and presence of lymphovascular invasion (LVI).Operation-related outcomes: type of vein resection/reconstruction performed, operative time, blood loss, rates of transfusion, status of resection margins (positive versus negative), rates of R0 and R1 resections, number of patients having venous infiltration at final histologic examination. Definition of resection margins varied among the included studies. Additionally, the definition of postoperative morbidity varied, although the main part of the complications were defined according to the International Study Group of Pancreatic Surgery [19,20,21].Duration of postoperative hospital stay.30-day mortality, overall and specific postoperative morbidity: pancreatic fistula, bile leak, delayed gastric emptying (DGE), hemorrhage, rates of blood transfusions, rates of reoperations, number of patients receiving adjuvant chemotherapy.Duration of follow-up and survival outcomes including 1-, 3-, and 5- year overall survival (OS) in both the PD+VR and PD groups. Survival was defined as the number of months between the date of surgery and death.

### 2.5. Statistical Analysis

Meta-analysis was performed in line with recommendations from the Preferred Reporting Items for Systematic Reviews and MetaAnalyses (PRISMA) statement [22]. RevMan software version 5.3 (The Cochrane Collaboration, Software Update, Oxford) was used to perform the meta-analysis. Variables were pooled only if evaluated by three or more studies. For dichotomous variables, odds ratios (ORs) were used as summary measures of efficacy, corresponding to the odds of an event occurring in the treatment group (PD+VR) compared to the reference group (PD).

An odd ratio of more than 1 indicates the probability that an outcome is more likely to occur in the first group, and it is considered statistically significant when P < 0.05 and when the 95% confidence interval (CI) does not include the value 1. The Mantel Haenszel method was used to combine the ORs for outcomes of interest. A random effect model, which is more robust in terms of anticipated heterogeneity, was used. The random effect-weighted mean difference (MD) between groups was used as the summary statistic for continuous variables; 95% confidence intervals were reported. For studies reporting medians and interquartile ranges instead of means and standard deviations, the method described by Hozo et al. was used to impute means and standard deviations. Statistical heterogeneity was evaluated using the I2 statistic. I2 values of 0 to 25%, 26% to 50%, and >51% were considered to be indicative of homogeneity, moderate heterogeneity, and high heterogeneity, respectively. All statistical data were considered significant if *p* < 0.05.

## 3. Results

### 3.1. Included Studies

The PRISMA flow diagram for systematic review is presented in Figure 1. Twenty-three studies comparing PD+VR and standard PD for patients with PHAC, published between May 1998 and November 2019, were considered suitable for the meta-analysis [8,12,23,24,25,26,27,28,29,30,31,32,33,34,35,36,37,38,39,40,41,42,43]. On review of the data extraction, there was 100% agreement between the three reviewers. Eight studies were conducted in USA, 4 studies were conducted in China, 3 studies were conducted in Japan, 2 studies were conducted in UK, 1 study was conducted in France, 1 study was conducted in Taiwan, 1 study was conducted in India, 1 study was conducted in South Korea, 1 study was conducted in Switzerland, and 1 study was conducted in Australia. The reports were primarily retrospective studies of comparable patients. No randomized trials were identified. A total of 6037 patients were included, of whom 1729 (28.6%) underwent PD+VR and 4308 (71.4%) underwent standard PD. Three studies only included patients having T3 PHAC [12,35,37]. The characteristics of the studies are summarized in Table 1. The results of the quality assessment of the studies, assessed using the Newcastle–Ottawa Scale scores for cohort studies, are summarized in Appendix A. Of the 23 studies, 22 were of high quality (scores between 6 and 8), and only one had a score of 5 points.

### 3.2. Patient and Tumor Characteristics

Patients in the two groups (PD+VR and PD) were similar with respect to age (mean difference [MD] 0.89, 95% CI -0.11 to 1.88, *p* = 0.08 ), ASA score (ASA I: OR 1.27, 95% CI 0.81 to 2.0, *p* = 0.30; ASA II: OR 0.75, 95% CI 0.53 to 1.05, *p* = 0.10; ASA III: OR 1.33, 95% CI 0.70 to 2.52, *p* = 0.38), and the need for preoperative biliary drainage (OR 0.91, 95% CI 0.61 to 1.35, *p* = 0.63), while rates of male gender were higher in the PD+VR group (OR 0.79, 95% CI 0.67 to 0.92, *p* = 0.003). Tumor size was significantly larger in the PD+VR group (MD 3.87, 95% CI 1.75 to 5.99, *p* = 0.0003), while the distribution of T1 and T2 categories did not differ (T1: OR 0.41, 95% CI 0.16 to 1.05, *p* = 0.06; T2: OR 0.30, 95% CI 0.05 to 1.67, *p* = 0.17). T3 tumors were more represented in the group PD+VR (OR 1.24, 95% CI 1.06 to 1.45, *p* = 0.007). Groups were similar with respect to tumor differentiation (well differentiated: OR 0.77, 95% CI 0.52 to 1.16, *p* = 0.22; moderately differentiated: OR 1.03, 95% CI 0.64 to 1.68, *p* = 0.90; poorly differentiated: OR 1.43, 95% CI 0.97 to 2.10, *p* = 0.07). However, when considering tumor stage, there were significantly more patients in the PD group in Stage 1 (OR 0.29, 95% CI 0.18 to 0.47, *p* < 0.00001), while there were significantly more patients in the PD+VR group in Stage 2 (OR 2.33, 95% CI 1.56 to 3.48, *p* < 0.0001). Lymphovascular invasion was slightly higher in the tumors of the PD+VR group, but not in a significant manner (OR 1.33, 95% CI 0.98 to 1.80, *p* = 0.07). Lymph node positivity was more frequently observed in the PD+VR group (OR 1.24, 95% CI 1.06 to 1.45, *p* = 0.007) (Appendix A). Nine studies reported on neoadjuvant chemotherapy. It was administrated in only in 5 of them [26,27,28,31,33,34,35,40,41]. In total, 473 patients received preoperative treatment: 266 in the PD+VR group and 207 in the PD group. Patients of the PD+VR group were more likely to receive neoadjuvant chemotherapy (*p* = 0.01) (Appendix A).

The type of venous resection/reconstruction performed was resumed in Appendix A.

### 3.3. Operative Outcomes

Blood loss was not statistically different in the two groups (MD 209.22, 95% CI −102.53 to 515.13, *p* = 0.16). Operative time was longer in the PD+VR group (MD 58.16, 95% CI 38.33 to 77.99, *p* < 0.0001). Rates of positive margins (OR 1.69, 95% CI 1.23 to 2.31, *p* = 0.001) and rates of R1 resections (OR 1.74, 95% CI 1.37 to 2.20, *p* < 0.0001) were higher in the PD+VR group, while R0 resections (OR 0.60, 95% CI 0.47 to 1.75, *p* < 0.0001), were in favor of the standard PD group (Figure 2).

### 3.4. Postoperative Outcomes

Rates of 30-day mortality (which also included data of in-hospital mortality of 3 studies) were higher in the PD+VR group (OR 1.93, 95% CI 1.28 to 2.91, *p* = 0.002). As for postoperative morbidity, there was no difference between the PD+VR and PD groups in rates of overall complications (OR, 1.07, 95% CI, 0.81 to 1.41, *p* = 0.65). Rates of pancreatic fistula were slightly higher in the standard PD group (OR 0.74, 95% CI 0.57 to 0.95, *p* = 0.02), while rates of bile leak (OR 1.65, 95% CI 0.39 to 7.00, *p* = 0.49), postoperative hemorrhage (OR 1.44, 95% CI 0.84 to 2.47, *p* = 0.19), and reoperation for postoperative complications (OR 1.21, 95% CI 0.78 to 1.87, *p* = 0.40) did not differ. Delayed gastric emptying (OR 1.56, 95% CI 1.19 to 2.05, *p* = 0.001) and the need for postoperative blood transfusions (OR 2.23, 95% CI 1.59 to 3.12, *p* < 0.0001) were lower in the standard PD group (Table 2). Hospital stay was slightly shorter in the PD+VR group, but the difference just fell short of statistical significance (MD −3.30, 95% CI −6.68 to 0.008, *p* = 0.06). (Figure 3). Only 65.8% (800/1216) of the patients belonging to the PD+VR group had a true histologic venous invasion by the tumor at final pathologic examination. Data on adjuvant treatments were available in 15 studies [12,23,24,26,27,28,29,30,31,32,34,35,38,41,43]. Among 3806 patients for whom this aspect was reported, 2220 (58.3%) received adjuvant chemotherapy or chemoradiation: 59.2% and 57.9% in the PD+VR and PD group, respectively (*p* = 0.19) (Appendix A).

### 3.5. Survival Outcomes

Survival outcomes were in favor of the standard PD group at 1-year OS (OR 0.79, 95% CI 0.67 to 0.92, *p* = 0.003), 3-year OS (OR 0.72, 95% CI 0.59 to 0.87, *p* = 0.0006), and 5-year OS (OR 0.57, 95% CI 0.39 to 0.83, *p* = 0.003) (Figure 4 and Table 2).

### 3.6. Assessment for Publication Bias

Funnel plots were constructed in order to assess for publication bias. The plots for the survival outcomes (Figure 5A–C) demonstrated some symmetry, as well as the plots for mortality, overall complications, pancreatic fistula, and reoperation (Figure 6A,B,D,E). In contrast, the plots for hemorrhage and transfusions (Figure 6C–F) suggested a decreased number of positive studies with low precision.

## 4. Discussion

In patients with borderline resectable PHAC, PD associated with portal–mesenteric resection is nowadays considered as a technically feasible option with acceptable morbidity and mortality rates [24,41,44,45]. It is noteworthy when considering that only 15–20% of patients with PHAC are candidates for radical resection at the time of diagnosis [8,12,42]. This evidence-based practice has dramatically changed the management of patients with borderline resectable carcinoma, as the addition of venous resection increases the number of patients candidates for tumor resection with curative intent [45].

However, data on survival benefits are conflicting in the available literature when PD+VR is compared with standard PD, since some studies have demonstrated worse survival with venous resection, while others have shown that it does not negatively affect survival after resection [5,15,44,46]. The main result of the present meta-analysis, where the inclusion criteria were restricted to patients undergoing PD with or without VR specifically for PHAC, is that 1-year, 3-year and 5-year overall survival rates were significantly lower after PD+VR. This is not surprising considering that the proportion of patients in that group had a more advanced disease at the time of surgery [6,12,23,34]. In fact, patients of the PD+VR group had a larger tumor size, a higher proportion of lymph node involvement, and more frequently belonged to Stage 2 than Stage 1. While some studies underscored that patients with vein involvement were more likely to have lymphovascular invasion, our pooled analysis failed to demonstrate any significant difference in this respect. During data extraction, we observed that the main part of the studies included in this meta-analysis reported that patients undergoing concurrent PD+VR can achieve long-term survival rates equivalent to those requiring standard PD. Xie et al. in a propensity score matching analysis showed that the survival of patients receiving PD+VR was even longer [42], as well as Howard et al., who reported a higher 1-year actual survival after PD+VR [29]. However, in the analysis of pooled data, survival rates still remained in favor of standard PD. When looking to other single-institution studies with the largest criteria, the results of the current meta-analysis are in line with them. For example, a large French multicentric population-based study demonstrated a poorer survival for those undergoing portal–venous resection, as did the currently largest single-center study from Japan [47,48]. In addition, previous systematic reviews and meta-analysis reporting in general results of pancreatic resections for cancer with “en bloc” venous resection showed worse 5-year survival rates in that subgroup [5,7]. 

Other than tumor characteristics, survival after surgical resection of PHAC is related to surgical margin status. We observed that rates of patients with infiltrate margins were significantly higher in patients undergoing PD+VR, although the definition of resection margin in PD was not homogeneous. The objective of PD+VR is to achieve a R0 resection, which ranges between 40% and 70% in all PD+VR cases described in the literature [8]. The differences can be partially explained by the technical level of the centers and the criteria of defining an R0 resection. Several studies have demonstrated that R0 resection is associated with survival benefit over R1 resection in patients with pancreatic adenocarcinoma [7,39,49,50]. In the present meta-analysis, rates of R1 resection were significantly higher in the PD+VR group, while those of R0 were higher in the PD group. One can argue that those findings may partially explain the difference in survival outcomes, although the definition of an R0 margin was not identical across the studies due to the fact that histopathologic reporting is not universally standardized. Recently, the International Study Group of Pancreatic Surgery suggested that venous resection should be attempted if R0 resection is feasible [51]. Since too few studies reported on differences in survival between patients obtaining an R0 resection, was not possible to make a pooled analysis on this item in the current meta-analysis.

To note, data on the use of preoperative chemotherapy or chemoradiation were available only in 5 out of the 23 studies included in our meta-analysis [31,33,34,38,40]. This low utilization of preoperative therapy is likely correlated to the fact that the main part of the studies was conducted before the year 2015, when upfront surgical resection was recommended in the presence of suspicious portal–mesenteric invasion. The role of preoperative chemotherapy in patients with borderline resectable PHAC has been gaining importance, as the potential advantages of obtaining a downstaging of the tumor and achieving free margins following PD cannot be overemphasized [7]. Moreover, the response to neoadjuvant therapy may help in identifying patients having tumors with more aggressive biological behavior and worse outcomes. Although the heterogeneity of the available trials reduces the strength of any conclusion, many reports demonstrated improved R0 resections and higher overall survival in patients receiving preoperative chemotherapy for resectable pancreatic cancer [52,53], including a recent randomized, multicenter phase 2/3 trial [54]. As a consequence, nowadays, there is a growing trend in the using of chemotherapy as upfront treatment in patients with borderline resectable tumors, although it is still unclear about whether this translates into increased cure rates [45,55]. For example, in the Dutch PREOPANC trial, preoperative chemoradiotherapy for resectable or borderline resectable pancreatic cancer did not show a significant overall survival benefit [56]. Various treatment regimens have been used, including FOLFIRINOX, gemcitabine-based induction chemotherapy followed by 5-FU, nab-paclitaxel plus gemcitabine, and capecitabine-based chemoradiation [9,35,53]. According to the NCCN guidelines, while there is limited evidence to recommend specific neoadjuvant regimens, most member institutions prefer neoadjuvant therapy and encourage participation in clinical trials at or coordinated through a high-volume center [9]. In addition, ESMO guidelines suggest that patients with borderline resectable tumors should be included in clinical trials. If this would not be the case, the best option might be upfront chemotherapy followed by chemoradiation and then surgery (recommendation of type IV,B) [53]. 

Another remarkable point is that only 65.8% of the patients had a histology-demonstrated invasion of the portal–mesenteric vein wall by tumor cells at final pathologic examination, among the 14 studies reporting on this parameter. Although in a recent multicentric cohort study, Ravikumar et al. [57] found that 26.5% of patients undergoing portal vein resection had histological evidence of vein involvement, the proportion of actual vein invasion in the literature ranges from 40% to 100% [58]. Ideally, only patients with true venous invasion should undergo PD+VR, even though the real impact of histologic venous invasion on survival rates remains a matter of debate [12,44]. Studies have shown that invasion of the wall of the veins, rather than adhesion, is associated with reduced survival [6,7,12]. Venous invasion might indicate aggressive tumor biology, but some authors did not find any statistically significant impact on life expectancy and median survival between patients who did and did not have histopathologic evidence of vein invasion [59,60].

In an era where the decision-making process in borderline resectable pancreatic tumors is experiencing profound changes, the issue of true venous invasion calls attention to the role of pre-therapeutic work up. Accurate staging with imaging is of the utmost importance because patients need to be stratified into potentially curative upfront surgery versus neaodjuvant treatments. The decision is based on subtle differences regarding the contour irregularity of the vein and/or its contact with the tumor seen on cross-sectional CT and MRI images [15,61,62]. Preoperative prediction of pathologic invasion of the portal–mesenteric wall is still limited, even with advanced imaging tools. Some recent studies have underscored the need for enhancement of pre-treatment investigations in patients with suspicious portal–mesenteric invasion and, generally speaking, with borderline resectable pancreatic cancer [12,31]. 

The available literature suggests that venous resection in combination with multimodality therapy may significantly improve long-term survival for patients having borderline pancreatic cancer [12,34,44]. Other than a low use of neoadjuvant therapy, we also noticed a suboptimal number of patients who underwent adjuvant treatment among the studies. In fact, only 58.3% received chemotherapy or chemoradiation following either PD or PD+VR. In this regard, we can speculate that many patients were unfit for postoperative chemotherapy, which is a common problem in the setting of patients who undergo surgery for pancreatic cancer.

Although in many of the comparative studies included in this meta-analysis, the overall mortality was not altered by venous resection [16,34,37,38], we found that 30-day mortality rates were significantly higher in the PD+VR group. Similar findings were also reported from Peng et al. [17], as well as from Giovinazzo et al, who reported that patients undergoing portal–mesenteric venous resection had increased mortality rates in a large meta-analysis evaluating all types of pancreatectomy (PD, total pancreatectomy, and distal pancreatectomy) [6]. These data could be explained by the presence of a more advanced disease and with the major technical difficulty of venous resection/reconstruction; in fact, the PD+VR group also required a longer operation time, although intraoperative blood loss did not differ in the current meta-analysis. In this regard, we also found that two potential variables affecting mortality rates, such as age and ASA classification were similar in the two groups. Concomitant morbidities might have had a role in the observed mortality rates, but unfortunately, that aspect was not reported in the studies included.

In general, the incidence of postoperative morbidity in patients receiving PD with or without venous resection remains relatively high with rates ranging between 20% and 50% [14,24]. To note, in the current study, postoperative complication rates, excluding mortality, did not statistically differ between the two groups, although they were slightly higher in the PD+VR group. This finding is in line with other single institutional studies and reviews on the topic, although the literature is conflicting. Two reports based on the American College of Surgeons National Surgical Quality Improvement Program (ACS-NSQIP) and National Inpatient Sample demonstrated a significant increase in overall morbidity in patients with vascular tumor involvement [63,64]. In the present meta-analysis, we found that rates of pancreatic fistula, which remains one of the most dreaded complication following PD [65,66,67], were less frequent in the PD+VR group. This finding has also been reported by other authors. A possible explanation would be that in patients with portal invasion, a fibrotic texture of the pancreatic remnant, and high incidence of obstructive pancreatitis with pancreatic duct dilatation due to the large size of the primary tumor, frequently occur. Those findings might decrease the formation of postoperative pancreatic fistula [5,68]. The need for postoperative blood transfusion was more frequently encountered in the PD+VR group, although the amount of intraoperative blood loss and the occurrence of postoperative hemorrhage were not different between the two groups. Delayed gastric emptying was more frequent in the PD+VR group. It is not simple to understand the causes behind this phenomenon, but perhaps the fact that not all the studies specified the proportion of patients receiving a pylorus-preserving PD may have influenced the pooled analysis.

With respect to postoperative morbidity, the two main results of our meta-analysis are that PD+VR is associated to higher rates of 30-day mortality and similar rates of total postoperative complications when compared with standard PD. These data are in agreement with those from another recent meta-analysis evaluating the safety and efficacy of PD+VR in comparison to PD for tumors located in the pancreatic head [17]. In general, association between low postoperative morbidity and high hospital volume suggests that PD+VR should be performed by trained surgical team with skills in vascular anastomosis practice, or in collaboration with a transplant or vascular surgeon [36].

The results of the present meta-analysis, along with the review of the current literature, allow us to make some considerations about portal–mesenteric resection in patients with PHAC. Long-term survival outcome after PD is related to several factors that remain to be better established; among them, the local extension of the primary tumor, lymph node infiltration status, tumor differentiation, and involvement of surgical margins seem to play a prominent role [15,16]. To date, it is not clear whether portal–venous infiltration is a consequence of local tumor progression or if it can be considered as an expression of aggressive tumor biology [15,23,25]. However, the current meta-analysis demonstrates that PD+VR does not improve patients’ long-term prognosis when compared with standard PD. Bell et al. [7] in a meta-analysis based on 16 studies published up to 2015 found that portal–superior mesenteric vein resection during PD was associated with a higher R1 rate, lower 5-year survival, and concluded that such an approach was not cost-effective. It is our view that the survival outcomes of patients receiving PD+VR should not only be compared with those of patients receiving standard PD, but rather with those who receive nonoperative or palliative treatments. 

As in any meta-analysis, our study is limited by the quality of studies included, none of which had a randomized design. In addition, despite measures taken to standardize surgical population and definitions, variations in inclusion criteria and operative technique between studies might have led to differences in outcomes. In particular, the techniques of vascular reconstruction included direct anastomosis, venorrhaphy/patch, and interposition graft, and to date, a consensus on the ideal method of vascular reconstruction is lacking [57]; these differences might have affected some of the results. Despite these acknowledged limitations, the present work represents a large comparative analysis of surgical and oncological outcomes in patients receiving venous resection for PHAC. In particular, this is the first systematic review and meta-analysis specifically addressing the effects of PD+VR only in patients having PHAC, which ensured the homogeneity of the research population and reduced bias. Furthermore, it attempts to critically evaluate the PD+VR procedure in the context of a multimodality treatment of pancreatic adenocarcinoma.

## 5. Conclusions

In conclusion, while the prognostic importance of venous invasion in patients undergoing PD+VR still needs to be established, the technique has significantly increased the resection rate in patients with PHAC. However, it should be taken into account that PD+VR, when compared with standard PD, has worse survival outcomes and higher 30-day mortality, whereas postoperative morbidity rates are similar. Further research studies are needed to evaluate the role of portal–mesenteric resection in the context of multimodality treatment of PHAC.

## Figures and Tables

**Figure 1 cancers-12-01976-f001:**
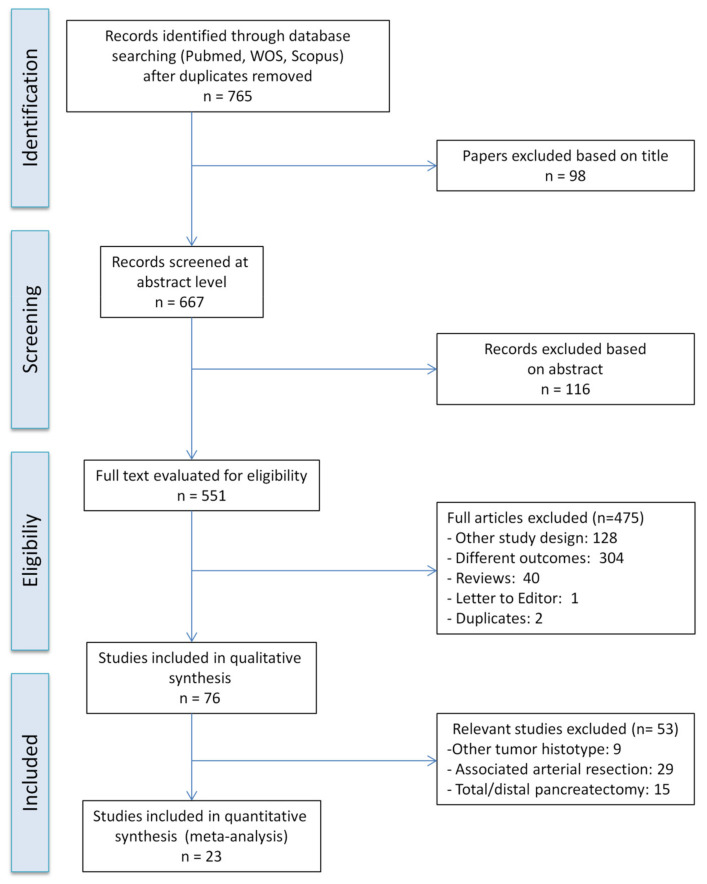
Flowchart of study selection for the meta-analysis.

**Figure 2 cancers-12-01976-f002:**
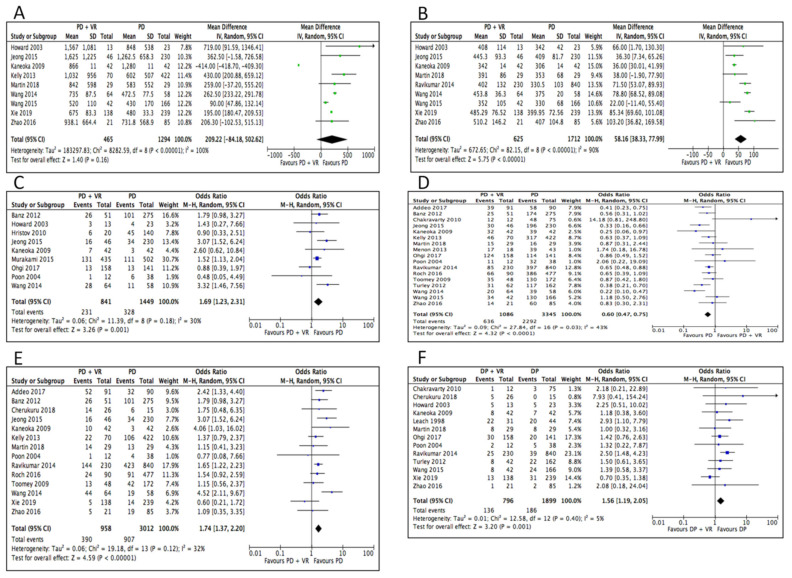
Meta-analysis of operative outcomes. (**A**) Blood loss. (**B**) Operative time. (**C**) Positive margin status. (**D**) R0 resection. (**E**) R1 resection. (**F**) Delayed gastric emptying.

**Figure 3 cancers-12-01976-f003:**
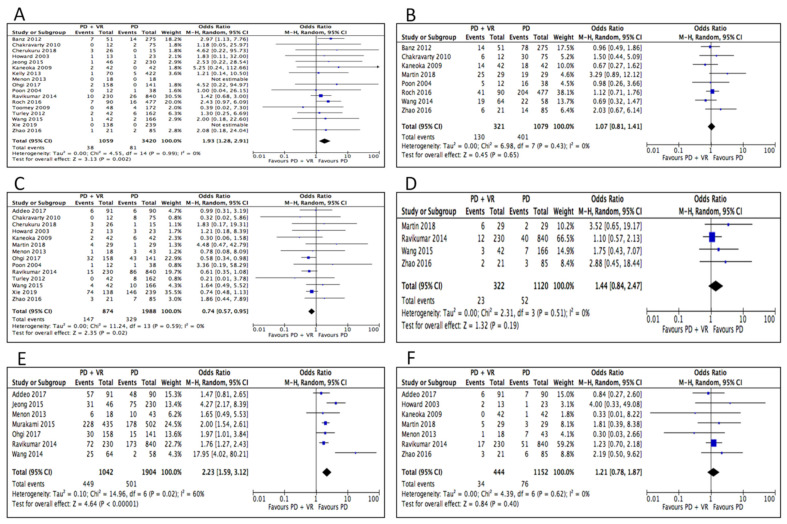
Meta-analysis of postoperative outcomes. (**A**) Mortality. (**B**) Overall complications. (**C**) Pancreatic fistula. (**D**) Hemorrage. (**E**) Transfusions. (**F**) Reoperation.

**Figure 4 cancers-12-01976-f004:**
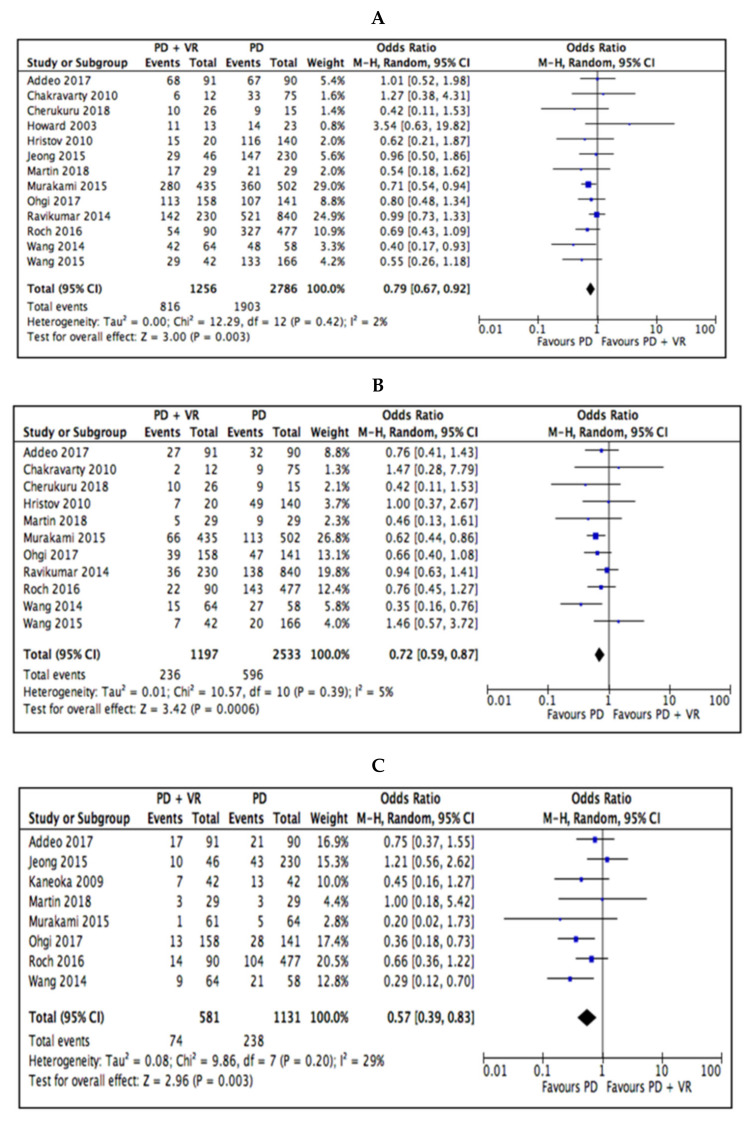
Meta-analysis of survival outcomes. (**A**) 1-year OS. (**B**) 3-year OS. (**C**) 5-year OS.

**Figure 5 cancers-12-01976-f005:**
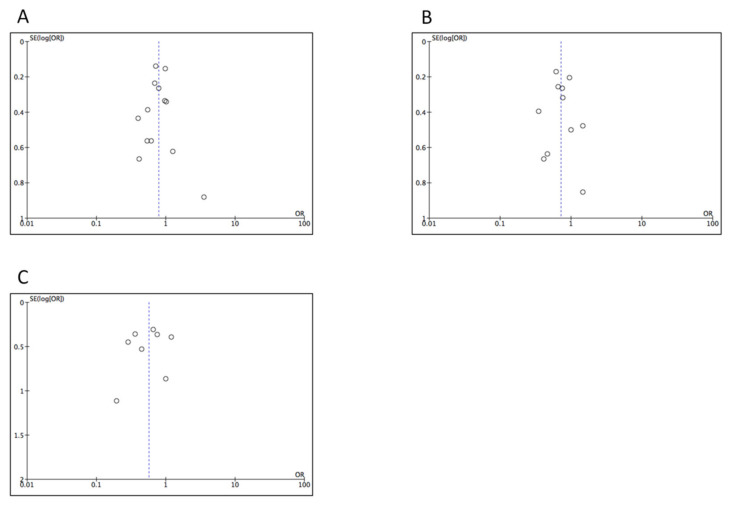
Funnel plot of survival outcomes (**A**) 1-year OS. (**B**) 3-year OS. (**C**) 5-year OS.

**Figure 6 cancers-12-01976-f006:**
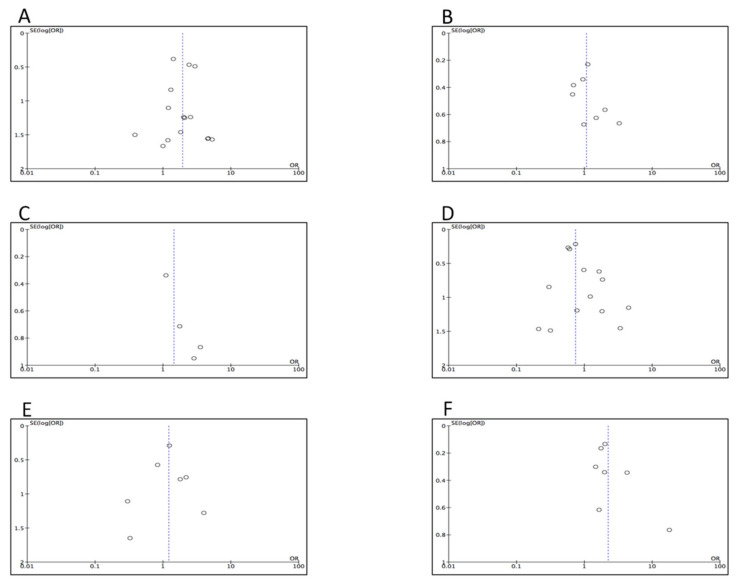
Funnel plots of postoperative outcomes. (**A**) Mortality. (**B**) Overall complications. (**C**) Hemorrage. (**D**) Pancreatic fistula. (**E**) Reoperation. (**F**) Transfusions.

**Table 1 cancers-12-01976-t001:** Characteristics of the studies included in the meta-analysis.

Author	Year	Country	Time Frame	PD+VR	PD	Age	Stage	Survival Outcomes
Addeo P [12]	2017	France	2006–2014	91	90	PD+VR: 66 PD: 67	NR	Median OS 27 vs. 22 months (*p* = 0.28)
Banz VM [23]	2011	UK	1997–2009	51	275	PD+VR: 67 PD 65:	IB: 3 (0.9%) IIA: 54 (16.6%) IIB: 268 (82.2%) IV: 1 (0.3%)	Median OS 14.8 vs. 14.5 months (*p* = 0.41)
Chakravarty K [24]	2010	Taiwan	1996–2006	12	75	PD+VR: 62.9 PD: 62.9	IIA: 14 (16.1%) IIB: 3 (3.4%)	5-year OS 16.7 vs. 12.2 (*p* = 0.9)
Cherukuru R [25]	2018	India	2010–2016	26	15	59 *	NR	Median OS 14 vs. 17 months (*p* = 0.9)
Howard TJ [29]	2003	USA	NR	13	23	PD+VR: 68 PD: 67.8	I: 6 (16.7%) II: 9 (25%) III: 21 (58.3%)	Median OS 12 vs. 13 months (*p* < 0.05)
Hristov B [28]	2010	USA	1993–2005	20	140	PD+VR: 63.5 PD: 54	NR	Median OS 20.8 vs. 21.4 months (*p* < 0.05)
Jeong J [30]	2015	South Korea	1995–2009	46	230	PD+VR: 60.5 PD: 61.5	NR	Median OS 16 vs. 12 months (*p* = 0.08)
Kaneoka Y [26]	2009	Japan	1993–2006	42	42	PD+VR: 66 PD: 65	IA: 6 (6.9%) IB: 6 (6.9%) IIA: 30 (34.5%) IIB: 38 (43.7%) III: 2 (2.3%) IV: 5 (5.7%)	Median OS 26 vs. 12 months (*p* = 0.04)
Kelly KJ [27]	2013	USA	2000–2007	70	422	PD+VR: 66.8 PD: 65	NR	Median OS 19.3 vs. 15.6 months (*p* = 0.05)
Leach SD [31]	1998	USA	1990–1995	31	44	PD+VR: 66 PD: 64	NR	Median OS 20 vs. 22 months (*p* = 0.25)
Martin D [32]	2018	Switzerland	2008–2013	29	29	PD+VR: 69 PD: 66	NR	Median OS 12.9 vs. 20.3 months (*p* = 0.13)
Menon VG [33]	2013	USA	2007–2012	18	43	PD+VR: 67.2 PR: 69	IIB: 41 (67.2%)	Median OS 31 vs. 31 months (*p* = 0.9)
Murakami Y [34]	2015	Japan	2001–2012	435	502	68 *	IA-I: 58 (6.2%) IIA-IIB-III-IV: 879 (93.8%)	Median OS 18.5 vs. 25.8 months (*p* < 0.01)
Ohgi K [35]	2017	Japan	2002–2014	158	141	PD+VR: 68 PD: 67	NR	Median OS 20.7 vs. 18.6 months (*p* = 0.8)
Poon RT [36]	2004	China	1998–2002	12	38	PD+VR: 61.5 PD: 62.5	I: 17 (34%) II: 7 (14%) III: 18 (36%) IV: 8 (16%)	Median OS 19.5 vs 20.7 months (*p* = 0.8)
Ravikumar R [37]	2014	UK	1998–2011	230	840	PD+VR: 65 PD: 66	NR	Median OS 18 vs. 18.2 months (*p* NR)
Roch AM [38]	2016	USA	2000–2014	90	477	PD+VR: 66.4 PD: 66.3	NR	Median OS 14 vs. 21 months (*p* = 0.08)
Toomey P [39]	2009	USA	NR	48	172	PD+VR: 67 PD: 68	IA: 10 (4.5%) IA: 40 (18.2%) IIA: 43 (19.6%) IIB: 127 (57.7%)	Median OS 18 vs. 17 months (*p* = 0.84)
Turley RS [40]	2012	USA	1997–2008	42	162	PD+VR: 63.5 PD: 66	NR	Median OS 21 vs. 20 months (*p* = 0.9)
Wang F [41]	2014	Australia	2004–2012	64	58	PD+VR: 66 PD: 67	IA: 1 (0.8%) IB: 3 (2.5%) IIA: 27 (22.5%) IIB: 75 (62.5%) III: 9 (7.5%) IV: 5 (4.2%)	Median OS 18 vs. 31 months (*p* = 0.02)
Wang WL [8]	2015	China	2009–2013	42	166	PD+VR: 59.4 PD: 60.5	NR	Median OS 20 vs 26 months (*p* = 0.2)
Xie ZB [42]	2019	China	2011–2013	138	239	PD+VR: 62.75 PD: 61.54	I: 136 (36.1%) II: 210 (55.7%) III: 31 (8.2%)	Median OS 25.1 vs. 29.3 months (*p* = 0.04)
Zhao X [43]	2016	China	2014–2016	21	85	PD+VR: 63 PD: 63.5	IA: 17 (16%) IB: 18 (17%) IIA: 6 (5.7%) IIB: 65 (61.3%)	Median OS 15 vs. 19 months (*p* < 0.05)

OS: overall survival; PDAC: pancreatic ductal adenocarcinoma; NR: not reported; PD: pancreaticoduodenectomy; PD+VR: pancreaticoduodenectomy + venous resection. * Mean age was reported for the entire cohort.

**Table 2 cancers-12-01976-t002:** Comparisons of postoperative outcomes and survival outcomes. LLCI: Lower Level Confidence Interval; ULCI: Upper Level Confidence Interval.

Variables	No. of Pooled Studies	Mean PD+VR	Mean PD	Difference or Odds Ratio	95% LLCI	95% ULCI
Blood loss (ml)	9	977.8	743.31	209.22	−84.18	502.62
Operative time (min)	9	421.1	361.4	58.16	38.3	77.9
Positive margin status	9	0.27	0.22	1.69	1.23	2.31
R0 resection	17	0.58	0.68	0.6	0.47	0.75
Delayed gastric emptying	13	0.17	0.09	1.56	1.19	2.05
Mortality	17	0.03	0.02	1.93	1.28	2.91
Overall complications	8	0.4	0.4	1.07	0.81	1.41
Pancreatic fistula	14	0.2	0.2	0.74	0.57	0.95
Postoperative Hemorrage	4	0.1	0.04	1.44	0.84	2.47
Reoperations	7	0.1	0.1	1.21	0.78	1.87
1-yr OS	13	0.6	0.7	0.79	0.67	0.92
3-yr OS	11	0.2	0.2	0.72	0.59	0.87
5-yr OS	8	0.1	0.2	0.57	0.39	0.83

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
