# Peer review of "The Impact on Survival and Morbidity of Portal–Mesenteric Resection During Pancreaticoduodenectomy for Pancreatic Head Adenocarcinoma: A Systematic Review and Meta-Analysis of Comparative Studies"

_cancers, 2020, doi:10.3390/cancers12071976_

Round 1

Reviewer 1 Report

Overall, this is a comprehensive and interesting manuscript. 

It would be useful to add the stage of the disease and age of patients in Table 1.

In addition please make clear if this meta-analysis refers to resectable or borderline tumors os both. 

Reviewer 2 Report

The authors present a systematic review and meta-analysis comparing pancreatoduodenectomy (PD) with and without portal-mesenteric vein resection for pancreatic head adenocarcinoma. The authors conclude that PD with venous resection (PD+VR) is feasible, but is associated with a higher rate of 30-day mortality and inferior survival outcomes compared with standard PD. I think that the paper is interesting and is methodologically sound (i.e. conducted according to the PRISMA guidelines, appropriate statistical analysis), but I have some comments to further improve the paper, which are listed below.

Title page:

  • Correspondence is mentioned twice on row 13.
  • It seems that two different fonts are used in the abstract.
  • Authors should be consistent with subheadings of the abstract, i.e. now only 'methods' is typed in bold.
  • Double spaces in lines 22, 24 and 31.
  • Please add keywords to line 34 and remove from the abstract.

Introduction:

  • Authors should carefully review the Introduction section for inconsistencies regarding references (i.e. double periods in line 39 and 47) and spelling errors (i.e. histoypes line 55).
  • Lines 55-58 are difficult to read and should be shortened. Also, survival outcomes are mentioned twice in this line.

Methods:

  • The authors should mention whether the used search terms were based on Medical Subject Headings (i.e. MeSH terms), title-abstract (tiab) or as free keywords. 
  • References including version should be added to the definitions section, i.e. which AJCC TNM-stage was used, which definitions were used to define post-operative complications (ISGPS?)
  • Authors should state whether survival was defined based on time from diagnosis or from surgery.
  • The full search strategies could be added as supplementary file.

Results:

  • Authors should review Figure 1 for spelling errors (i.e. reviewies).
  • Instead of copying the conclusions of the included articles in Table 1, raw data of the included studies should be presented. The authors could also consider adding a second table including overall survival, R0-resection rate, tumor characteristics, complications etc. stratified by PD/PD+VR per study.
  • Five of the included studies reported on the use of neoadjuvant chemotherapy. The authors could perform a sensitivity analysis to compare outcomes between PD and PD+VR after neoadjuvant chemotherapy.
  • Likewise, I encourage the authors to perform a sensitivity analysis comparing patients who received adjuvant chemotherapy. 
  • Type of venous resection (i.e. wedge or segment resection, type of reconstruction, use of grafts etc.) of the included studies should be reported. Was there a difference in outcomes between different types of venous resection? 

Discussion

  • Authors should carefully review the Discussion section for spelling errors, i.e. NNCC guidelines line 279, FOLFORINOX line 278.
  • Recently, two randomized-trials were published comparing upfront surgery with neoadjuvant chemoradiation followed by surgery for (borderline) resectable pancreatic cancer (Versteijne et al. JCO 2020), Jang et al. Ann Surg 2018). The authors could reflect on the results of these trials regarding the benefit of neoadjuvant therapy in pancreatic cancer.

Supplements:

  • Total NOS-score should be added to Supplementary Table A, to allow for easier comparison between studies.

Round 2

Reviewer 2 Report

The authors have improved their manuscript and added additional sensitivity analyses regarding adjuvant and neoadjuvant chemotherapy. In addition, tables were added comparing surgical and oncological outcomes between studies. I have no further comments and recommend the journal to accept the paper.